# Investigating the Ovarian Microstructure in the Genera *Helicolenus* and *Scorpaena* (Teleostei, Sub-Order Scorpaenoidei) with Implications for Ovarian Dynamics and Spawning

**DOI:** 10.3390/ani12111412

**Published:** 2022-05-30

**Authors:** Cristina Porcu, Eleonora Lai, Andrea Bellodi, Pierluigi Carbonara, Alessandro Cau, Antonello Mulas, Noemi Pascale, Riccardo Porceddu, Maria Cristina Follesa

**Affiliations:** 1Dipartimento di Scienze della Vita e dell’Ambiente, Università degli Studi di Cagliari, Via Tommaso Fiorelli 1, 09126 Cagliari, Italy; cporcu@unica.it (C.P.); elelai93@gmail.com (E.L.); abellodi@unica.it (A.B.); alessandrocau@unica.it (A.C.); amulas@unica.it (A.M.); pascalenoemi3@gmail.com (N.P.); riccardo.porceddu@unica.it (R.P.); 2Consorzio Nazionale Interuniversitario per le Scienze Mare (CoNISMa), Piazzale Flaminio 9, 00196 Roma, Italy; 3COISPA Tecnologia & Ricerca, Stazione Sperimentale per lo Studio delle Risorse del Mare, Via dei Trulli 18, 70126 Bari, Italy; carbonara@coispa.it

**Keywords:** rockfish, gonad histology, morphology, ovarian dynamic, Central–Western Mediterranean

## Abstract

**Simple Summary:**

The diversity of reproductive mechanism in bony fishes is greater than in any other group of vertebrates. It ranges from oviparous, to several stages of viviparous forms. In this context, scorpaenoid fishes belonging to the families Scorpaenidae and Sebastidae are of particular interest, since they show extremely varied reproductive modes connected with ovarian structures. We describe here the ovarian morphology of five rockfish species showing different reproductive modalities, using histology. Specialized microscopic features were found during gametogenesis, strictly related to the production of gelatinous mass surrounding the eggs, typical of these species. Based on microscopic maturity stages here analyzed, we found that all species shed eggs more than once through the spawning season, and were characterized by continuous oogenesis with multiple oocyte deposition. Further ovarian dynamic observations supported the hypothesis that all species had an indeterminate fecundity.

**Abstract:**

The sub-order Scorpenoidei appears to be particularly interesting due to the presence of intermediate stages between oviparity and viviparity in several species. The present study aims to describe the ovarian morphology, using a histological and histochemical approach, in four ovuliparous species belonging to *Scorpaena* genus compared with a zygoparous species, *H. dactylopterus*, focusing also on the assessment of the ovarian dynamics in the populations of such species in Sardinia waters (central–western Mediterranean). Ovarian sections of all species were examined using light microscopy. All species showed a specialized ovary, cystovarian type II-3, strictly related to the production of gelatinous matrices surrounding the eggs. Some microscopic peculiarities in the oogenesis process were found: thin zona pellucida, small and low cortical alveoli, and a specialized ovarian wall during the spawning period. All species analyzed were batch-spawners with an asynchronous ovarian organization. A continuous recruitment of oocytes and the occurrence of de novo vitellogenesis was also observed. During the spawning period, low atresia intensity was detected, while a marked increase in this intensity found in the ovaries at the end of spawning season. Our observations may support an indeterminate fecundity type for these species.

## 1. Introduction

The diversity of reproductive mechanisms in teleost fishes is greater than in any other group of vertebrates. It ranges from oviparous, through several stages of lecithotrophic viviparity, to highly matrotrophic viviparous forms [1]. In this context, scorpaenoid fish (Scorpaenoidei), belonging to the families Scorpaenidae and Sebastidae, are of particular interest, since they show extremely varied reproductive modes [2,3]. Oviparity is common to most genera, and includes three levels: ovuliparity, zygoparity, and embryoparity [4,5]. Ovuliparity refers to the release of eggs from females into the water column, which are then fertilized by males (external insemination), and is known in the genera *Pontinus*, *Pterois*, *Scorpaena*, *Scorpaenopsis*, *Sebastapistes*, and *Scorpaenodes* (e.g., [6,7]). Zygoparity refers to an oviparous reproductive pattern in which the fertilization is internal, and early developed embryos are released into the environment after a short period of time. It is known only in the bluemouth *Helicolenus dactylopterus* (Delaroche, 1890) (family Sebastidae) [8,9], where the female is able to store sperm in the ovary for a long time, within specialized structures where spermatozoa are maintained in a viable state until oocyte maturation (e.g., [10,11,12,13]). Sperm is then released towards the ovarian lumen to fertilize the mature oocytes. More specifically, in *H. dactylopterus*, most of the embryos are in the early-celled stage, whereas blastula and tail bud stages only represent a small percentage [14,15]. Embryoparity is an oviparous mode in which the embryo is formed, the period of retention after fertilization is prolonged, and embryos may develop within the maternal body to quite an advanced state prior to their release (genus *Sebastolobus*, [7]). Consequently, the extreme limits of embryoparity can overlap with those of viviparity (i.e., free embryos or larvae released in the environment), observed in the genera *Sebastes* and *Sebasticus* and in the species *Helicolenus percoides* (e.g., [6,7,16]).

The ovaries of teleosts can be generally classified into two major types according to their structures: gymnovarian and cystovarian types [17]. In the first type, the ovary is open to the abdominal cavity, the ovarian duct is absent, and their oocytes are released directly into the coelomic cavity and carried to the outside through their genital papilla (e.g., [18]). In the second, the ovary is enclosed with a muscular or non-muscular ovarian membrane, and the ovarian duct is present, leading oocytes to the external environment. The cystovarian type is further divided into four types based on histological characteristics: II-1, II-2, II-3, and II-4 [19]. Generally, the ovarian type corresponds to the phylogenetic classification comparatively well, and most species belonging to the same family have the same type of ovary [6]. Overall, two ovarian variations have been identified in the Scorpaenoidei [19]: (1) the ovary in which the lamella-like stroma develops from the ovarian hilus located on the dorsal side, and the ovarian cavity is located on the ventral side of the ovary, classified as cystovarian type II-1 (type II-1); (2) the ovary in which the stroma develops radially around the blood circulatory system that traverses the center of the ovary, and the ovarian cavity surrounds all the components around the ovary, classified as cystovarian type II-3 (type II-3) [19]. It seems that there is a connection between these two ovarian structures and the reproductive mode of scorpaenids [6]. Up until the present time, the type II-1 ovary has been found in two viviparous genera, while the type II-3 ovary has been identified in six oviparous ones (ovulipari) (e.g., [11,20]) and in *Helicolenus* genus (zygoparous), but it is uncertain whether it can be found also in other scorpaenid species. It is almost certain that the cystovarian type II-3 is an important structure able to form a gelatinous egg mass specific for scorpaenids [6]. Several characteristics distinguish many scorpaenids from the simpler ovuliparous species widely described in literature [3], but they are not described in all species.

The sub-order Scorpenoidei is represented in the Mediterranean Sea by two families [21,22]: Scorpaenidae and Sebastidae. Both families include two sub-families: Scorpaeninae and Pteroninae, and Sebastinae and Sebastolobinae, respectively. The Scorpeninae sub-family includes three genera (*Pontinus*, *Scorpaenodes,* and *Scorpaena*) and nine species, while the *Pteroninae* subfamily comprises a single genus and one species (*Pterois miles*). The Sebastinae and Sebastolobinae sub-families are both represented by one genus and one single species: *Helicolenus dactylopterus* and *Trachyscopia cristulata 3evelop3*, respectively [21,22]. All of these families represent an important component of commercial and recreational fisheries distributed throughout the Mediterranean Sea. In particular, rockfish, belonging to the genus *Scorpaena*, are one of the most important and valuable captures for fisheries in coastal areas around the Mediterranean Sea (e.g., [23,24,25,26,27,28]), and are mainly represented by the black scorpionfish *Scorpaena porcus* Linnaeus, 1758, the small red scorpionfish *S. notata* Rafinesque, 1810, and the red scorpionfish *S. scrofa* Linnaeus, 1758. The Sebastidae *H. dactylopterus*, instead, is a benthic deep-water species caught mainly by bottom trawls [29,30], playing an important ecological role in deep-sea fish communities [31], and is exploited in deep-sea fisheries targeted at deep-water crustaceans [30].

Given its particular reproductive mode, several studies on the reproduction of *H. dactylopterus* have been published (e.g., [8,10,15,32,33,34]), while this kind of information is still scarce for Mediterranean species belonging to the genus *Scorpaena* [11,35,36,37]. 

For this reason, the present study aims to describe the ovarian morphology with a histological and histochemical approach, including composition, morphology, and morphometry of oocytes in four ovuliparous Mediterranean rockfish species (*Scorpaena 3evelop3, S. notata*, *S. porcus,* and *S. scrofa*) in comparison with the same data obtained from the Sebastidae *H. dactylopterus*, zygoparous. Since many characteristics of the reproductive biology affecting their reproductive potential are unknown, this study is also focused in assessing the ovarian dynamic in these species populations in central–western Mediterranean examining, through histological analysis, the development pattern and the growth of multiple oocyte groups and, finally, quantifying the atresia rate of the final number of developed oocytes, which is useful for determining the oocyte recruitment process of these species.

## 2. Materials and Methods

### 2.1. Sample Collection

Female specimens of four species belonging to the Scorpenidae family (*Scorpaena elongata*, *S. notata*, *S. porcus,* and *S. scrofa*) and one Sebastidae species (*Helicolenus dactylopterus*) were collected around Sardinian waters (central–western Mediterranean) during the Mediterranean International Trawl Survey (MEDITS; [38]), along with data collected monthly from commercial landings through the Data Collection Framework (European Union Regulation 199/2008). *H. dactylopterus* and *S. elongata* were sampled exclusively from bottom trawls, while *S. notata*, *S. porcus* and *S. scrofa* were sampled from both bottom trawls and trammel nets (Table 1). The collection and handling of animals strictly followed the ethical and welfare considerations approved by the ethics committee of the University of Cagliari (Sardinia, Italy).

For each specimen, the following parameters were measured: total length (TL, cm) to the nearest half cm; and total weight (TW, g) to the nearest 0.01 g. Sex and ovary maturation were recorded. The maturity status was assessed by dissection according to macroscopic criteria established by [39,40], considering the dimension of the ovaries with respect to the celomatic cavity, their degree of opacity, consistency and vascularization, oocyte visibility, and overall coloration. All females were classified in seven stages as follows: 1, immature virgin; 2A, developing virgin, 2B, recovering; 2C, maturing; 3, mature/spawner; 4A, spent; 4B, resting. In *H. dactylopterus*, the stage 3 (mature/spawner) was modified ad hoc, and subdivided in 3a (absence of embryos) and 3b (presence of embryos).

Ovaries were removed, weighed (OW, 0.1 g), and preserved in 5% buffered formalin for histological examination (pH 7.4, 0.1 M).

### 2.2. Histological Preparation

A small piece of ovarian tissue (0.5 to 1 cm long) was processed for histological analysis. The tissues were dehydrated and embedded in a synthetic resin (GMA, Technovit 7100, Bio-Optica, Milan, Italy) following routine protocols, and sectioned at 3.5 µm with a rotating microtome (ARM3750, Histo-Line Laboratories, Pantigliate, Italy). Slides were stained with Gill hematoxylin, followed by eosin counterstain (H&E) for standard histology, and with periodic acid–Schiff (PAS) and Alcian blue (AB) in combination to assess the production of neutral and sulfated acid mucins [41]. Subsequently, sections were dehydrated in graded ethanol (96–100%), cleared in Histolemon (Carlo Erba Reagents, Cornaredo, Italy), and mounted in resin (Eukitt, Bio-Optica, Milano, Italy).

### 2.3. Ovarian Dynamics 

Histological samples were examined to determine the developmental stage of oocytes, the presence/absence of post-ovulatory follicles (POFs), and the prevalence and intensity of atretic oocytes. Oocytes were assigned to stages in accordance with [42]; from least to most developed stage, included primary growth (PG) and multiple secondary growth (SG) stages consisting of: cortical alveoli (CA), primary vitellogenic (Vtg1, small granules of yolk that first appear around either the periphery of the oocyte or the nucleus), secondary vitellogenic (Vtg2, larger yolk globules throughout the cytoplasm.), tertiary vitellogenic (Vtg3, numerous large yolk globules fill the cytoplasm, and oil droplets, if present, begin to surround the nucleus), germinal migration (GVM), and hydration (H) stages. POFs were classified based on their degree of degeneration: those having bigger size and thicker and more convoluted epithelium composed of linearly arranged granulosa cells were classified as new, whilst those with signs of degeneration were classified as old. 

Concerning atresia, females were assigned to atretic states 0, 1, and 2, having 0%, <50%, and ≥50% SG oocytes with α-atresia, respectively [43]. All of these histological markers were used to assign females to different spawning subphases [42]. In particular, actively spawning (AS) females were categorized as those displaying markers of imminent or recent spawning activity such as GVM, H, or POFs, while non-spawning (NS) was the category assigned to all remaining SG females with no spawning markers.

The size composition of oocytes was obtained measuring only oocytes where the nucleus was clear, and given that these oocytes are rarely perfectly spherical in shape, in order to reduce the variance, the diameter of each oocyte was taken using TpsDig software [44], and calculated as average of the major and minor axis [45,46,47]. For each species, at the maturing and mature/spawner maturity stages, the thickness of zona pellucida (µm) of oocytes (*n* = 35 for each stage) from the cortical alveoli stage to the germinal migration stage was taken. Differences in mean oocyte diameters and zona pellucida thickness among maturity stages were tested using one-way ANOVA [48].

Ovarian dynamics were assessed in a subsample of females in each species. Oocytes in each subsample were grouped into size classes of 50 µm in order to characterize the size cohorts (oocyte size frequency distribution size, OSFD).

The relative intensity of atresia (RIA), i.e., the percentage of α- atretic oocytes in relation to the total SG oocytes in an individual ovary, was also recorded from histological sections of females. The co-occurrence of different spawning markers (e.g., POFs with GVM oocytes) was utilized as an index of spawning interval, SI.

The seasonality of spawning was estimated for *S. scrofa*, *S. porcus* and *H. dactylopterus* through an analysis of the seasonal or monthly distribution of the percentage of maturity stages of females, as well as the evolution of the mean gonadosomatic index (GSI) estimated per sampling date and maturity stage, and calculated using the ovary free body weight (OFW = TW − OW) [49]:GSI = 100 × OW × (OFW)^−1^(1)

## 3. Results

### 3.1. Ovarian Morphology

All species analyzed showed paired, saccular ovaries, entirely separated from each other, laying parallel in the dorsal part of the peritoneal cavity, and fused just before the genital opening (Figure 1A–E in detail). The ovary was classified as cystovarian type II-3 (type II-3), in which the stroma develops radially around the blood circulatory system, traversing the center of the ovary, and the ovarian cavity surrounds all components around the ovary. The connective tissue, including the germ cells (oocytes), extends radially from the central ovarian hilus to the surrounding ovarian wall, forming central stroma (Figure 1). The ovarian lamellae are suspended from the rachis by a fibromuscular trunk which contains blood vessels, and which has the surface covered with oocytes in different stages of maturity. As the oocytes develop, they take up a position further away from the rachis and closer to the ovarian lumen, so that the different stages are lined up in order of development (Figure 1A–E). During the mature phase, eggs are embedded in a large, pelagic gelatinous matrix in all species analyzed.

Holocrine glands, reacting positively to PAS staining, are present in all species analyzed (Figure 2A,B) (Table 2). The ovarian follicles are connected to the stroma by a narrow, vascularized peduncle, formed by a smooth and mono-stratified lamellar epithelium, the length of which increases as vitellogenesis advances (Figure 2C,D). Mature oocytes containing PAS positive ovarian fluid, which is particularly abundant during the spawning period, are expelled into the lumen. The zona pellucida (Figure 2E,F), strongly PAS-positive, is visible from the cortical alveoli stage (CA), and become thicker with the proceeding of the maturation phase, reaching maximum dimensions during the GVM stage, being made up of three layers, in all studied species (Figure 3). The thin outermost and the intermediate, medium electron dense region form the outer zona pellucida, while the internal region is a multilamellar striated region (inner zona pellucida) (Figure 2E–G).

The ovarian wall is made up of three layers (Figure 2H,I). The outer layer is composed by a mesothelium and the middle layer has two sub-layers (an outer one formed by a circular muscular system, and an inner one by much thicker longitudinal muscular system), which contains smooth musculature and numerous blood vessels (Figure 2H). The innermost layer (internal epithelium) of the ovary is covered with a simple cuboidal epithelium, and, during the period of vitellogenesis and spawning, modifies its structure, containing strongly PAS positive granules and cytoplasmic projections (Figure 2I) (Table 2).

In the zygoparous *H. dactylopterus*, cryptal structures packed with spermatozoa (here called spermatic crypts) near the basal part of the stroma are observed (Figure 2J). During the storage period, the cryptal epithelium releases strongly PAS-positive granules (Figure 2J).

### 3.2. Ovarian Dynamics

The progressive change in cellular diameters of the five studied species is shown in Figure 4. Differences among developmental maturity stages were always statistically significant (ANOVA, *p* < 0.001). *H. dactylopterus* showed the largest oocyte dimensions, followed by *S. scrofa* (Figure 4A,E), while *S. notata* the smallest one (Figure 4B).

Photo identification of the microscopic female maturity scale for the five species studied is presented in Figure 5. 

Females in PG phase showed that only PG oocytes through the perinucleolar stage in the ovaries (immature stages). As females move into secondary growth phase, they can be histologically distinguished by the initial appearance of CA oocytes (recovering stage, 2b). In these species, CA oocytes were characterized by very small and low in number cortical alveoli (weakly PAS positive), as well as a lack of oil droplets. Females in the maturing stage (stage 2C, NS) were distinguishable for the appearance of Vtg1 and Vtg2 oocytes, with no evidence of POFs or Vtg3 oocytes. Vitellogenic oocytes observed in species belonging to the genus *Scorpaena* were characterized by the complete absence of oil droplets in the cytoplasm, which, instead, were visible in *H. dactylopterus* oocytes (Figure 5E). Actively spawning females (AS, stage mature/spawner) were characterized by the presence of the previous described stages, and Vtg3 and the early stages of oocyte maturation (such as GVM oocytes), hydrated oocytes present simultaneously and POFs (Figure 6A–C). In *H. dactylopterus*, the presence of embryos in the gelatinous mass were recognizable with the appearance of hydrated oocytes, but with an undifferentiated mass of cells. In the post-spawner phase (PS, spent/resting), females showed a great amount of atresia of vitellogenic oocytes and PG oocytes (Figure 5 and Figure 6D–F).

The evolution of the percentage of the ovarian subphases above described (Figure 5), together with the trend of the seasonal Gonado somatic index (GSI) (Figure 7A,B) showed the presence of immature *S. scrofa* and *S. porcus* females with exclusively PG oocytes occurred in winter months. The proportion of SG females started in spring, where the most part of them were at the CA stage, while the remaining were at various stages of ovarian developmental, but not exhibited spawning markers, and were thus identified as NS. AS females occurred during the summer months, as confirmed by the high mean value of GSI (AS *S. scrofa* females, GSI 1.8–11.20%; AS *S. porcus* females, GSI 1.11–14.18%). PS females were recorded in low proportion in summer and in the autumn (mainly in October) (Figure 7A,B).

In *H. dactylopterus*, AS females with embryos in early-celled developmental stage with an intraovarian gelatinous matrix (GSI 1.18–9.48%) were observed from November to March (Figure 4E and Figure 5E), with high percentages in January (Figure 7C). From April to October, PG and NS females were observed in different percentages (Figure 7C). Moreover, spermatic crypts were observed in females from August (at the recovering stage, NS) to March (at spent stage, PS). 

The low number of *S. elongata* and *S. notata* females studied did not permit an estimation of a seasonality of spawning. However, *S. elongata* AS females (GSI 2.54–3.76%) were observed in July and September, while *S. notata* AS females (GSI 2.04–15.34%) in August and September.

The oocyte size frequency distributions (OSFD) from subsamples of mature/spawner females (AS) of the selected species are presented in Figure 8. During the seasonality of spawning, several cohorts of oocyte in all development stages (see oocyte’s dimension in Figure 4) were observed in all species, highlighting an asynchronous organization of the ovaries (Figure 8A–E). The OSFD was continuous, without any evident hiatus between primary and secondary growth stage oocytes, with PG representing always the most abundant oocytes. CA oocytes, although represented a small fraction of the total number of the oocyte population in all species, were always present in all females analyzed, as well as the Vtg1 oocytes. When spawning has started and at least one batch has already been released (ovaries with POFs), the OSFD showed several cohorts consisting of oocytes in early (Vtg1) and advanced vitellogenesis (including GVM oocytes).

In *H. dactylopterus* and in the species belonging to the genus *Scorpaena*, during the seasonality of spawning, the new POFs (formed few hours before) co-occurred with GVM oocytes in the late stage of development (to be spawned in the next hours) (Figure 6A,B), providing a strong indication that SI (spawning interval) should not exceed two days.

Regarding the relative intensity of atresia (RIA), it was estimated only in the species for which the spawning period was established (*S. scrofa*, *S. porcus,* and *H. dactylopterus*) (Figure 9). SG oocytes with α-atresia were always present through the spawning period in *S. scrofa* and *S. porcus* (Figure 9A), while in *H. dactylopterus,* no α-atresia (Atretic 0) was observed in November (at the beginning of reproductive period) (Figure 9B). However, atresia phenomenon was always relatively low (Atretic 1), with a strong increase towards the end of spawning, mainly in September in *S. scrofa* and *S. porcus*, (Figure 8A), and in March in *H. dactylopterus* (Figure 8B).

## 4. Discussion

The ovarian structure of Scorpenoid fishes analyzed in this paper differs from that observed in the majority of teleosts. The cystovarian type II-3, indeed, represents a specialized structure strictly related to the production of gelatinous matrices that surround the eggs, characteristics of the Scorpaenidae and Sebastidae families (e.g., [6]). Comparing the histological sections of the *Scorpaena* species ovaries with those of *H. dactylopterus*, indeed, many features resulted to be similar involved in facilitating the production of gelatinous egg masses.

In each studied species, the oocytes (starting from those at cortical alveoli stage) are supplied by a highly vascularized peduncle, which become longer with the increase of the oocyte diameter. This specialization is found in viviparous [50] and oviparous species [51], as well as some other vertebrates, including birds and reptiles [52]. Several functions are attributed to this structure. In ovuliparous oviparous species, such as the four studied species of the genus *Scorpaena*, peduncles have the main function to preventing oocyte crowding [11,53] facilitating the ovulation of mature oocytes in the gelatinous masses, while, in species with internal fertilization such as *H. dactylopterus*, the functions of peduncle are to provide access of spermatozoa to the oocytes during fertilization and access to nutrients for the embryos [11,54].

In the species of the genus *Scorpaena*, here analyzed, despite the pelagic nature of the eggs, the number of cortical alveoli is low and oil droplets are totally lacking also in vitellogenic oocytes, making the swelling of the eggs almost absent after the cortical reaction. Similar features have been registered in the oocytes of other species of the genera *Scorpaena* (not investigated in this paper) [55,56], and of the genera *Scorpaenopsis* and *Sebastapistes* [57,58], while in the ovuliparous species belonging to *Pterois* genus, a proliferation of cortical alveoli was displaced, together with a considerable deposition of lipid droplets occurred in the vitellogenic oocytes [20]. A phenomenon similar to what was reported for the *Pterois* genus has also been registered in the Sebastidae *H. dactylopterus*, where cortical alveoli are small and low, but lipid droplets began to accumulate in the oocyte cytoplasm within the secondary growth phase at the same time of cortical alveoli precursors, confirming the pelagic phase of the eggs [11].

Another peculiarity in the process of oogenesis in the studied species, strictly related with viviparity (e.g., [59,60,61,62]), is the thickness of the zona pellucida (zp). *S. scrofa* and *S. elongata* showed a maximum thickness of zona pellucida of 5.8 µm, while the zygoparous *H. dactylopterus* a maximum value of 6.2 µm, which are considerably less than that found in other related oviparous species, such as *Trigla lyra* (zp, 19 µm) and *Chelidonichthys obscurus* (zp, 37 µm) [63], but larger than those found in other Scorpaenidae species, such as *Dendrochirus zebra*, zp, 1 µm; *Scorpaenopsis possi*, zp, 1.4 µm; *Sebastapistes cyanostigma*, zp, 0.7 µm [7,64]. The zona pellucida of teleosts oocytes is complex, usually consisting of layers crossed by pores or channels [63], the morphology of which varies between species [64]. It is interrupted at the animal pole region by a specialized opening, the micropyle, which allows the passage of sperm in fertilization [65]. Zona pellucida plays various roles during oogenesis, egg deposition, fertilization, and embryogenesis. It is involved not only in the nourishment of the theca and granulosa layers and of the embryo, but also in the secretion of enzymes, the transportation of yolk material in early developmental stages, and the fixation of a deposited egg to the substratum. It is also implicated in the sperm attraction and prevention of polyspermy and in the antibacterial and mechanical protection during spawning, fertilization, and post fertilization periods Selman [63]. For the developing embryo, the egg envelope enables gas exchange, excretion, and transport of nutrients from the external environment [66].

As a general rule, viviparous fishes are characterized by a very thin zona pellucida, as recorded in viviparous scorpaenoids, such as *Sebastes paucispinis,* in which the zona pellucida was 1 µm thick [67,68]. If, in *H. dactylopterus*, the thickness of zona pellucida can be justified with its reproductive mode (e.g., nourishment of embryos), in a species of the genus *Scorpaena*, the reduction of thickness could imply loss of mechanical protection (essential when spawning is pelagic), which should be alternatively provided by the gelatinous matrix that encloses the eggs, as also reported by [11] in *S. notata*. In addition, as reported in other Scorpenoidei [11], all analyzed species showed during the spawning period a specialization and modification of the ovarian wall with internal epithelium secreting polysaccharides (PAS+) and highly developed cytoplasmatic projections, which are probably related to the great production of the ovarian fluid (always PAS+) present in the ovary. This type of secretory activity also found in the holocrine glands of the stroma could be responsible for the production of ovarian fluid, as already observed in *S. notata* [11].

The structuration of the spermatic crypts shown in *H. dactylopterus* is surely more complex than those described in viviparous species belonging to the same family, in which free swimming sperm is present in the ovarian fluid or singly adhered to the ovarian epithelium (e.g., [69,70]). Only recently, in the viviparous *Sebastes schelegelii*, swimming spermatozoa were found in the ovarian fluid at the early storage stage, while most spermatozoa were wrapped in the crypt structures between the multi-layered columnar epithelium and follicular layer of oocytes [16].

The mean monthly values of gonadosomatic index and the relative fractions of various developmental stages and spawning phases allowed for an estimation of a reliable spawning seasonality of *S. scrofa*, *S. porcus,* and *H. dactylopterus* in Sardinian waters (central–western Mediterranean Sea), until now unknown. A narrow reproductive period was found in the two species of *Scorpaena*. Actively spawning females of *S. scrofa* were observed mainly during the summer months from July to September, as well as females of *S. porcus*. These data are consistent with the few observations available for the Mediterranean Sea, which reported a very narrow spawning season (in summer) for both species (for *S. scrofa*: [36,69]; for *S. porcus*: [37,71,72]). The few data here provided for *S. notata* (mature females in August and September) were comparable with the only reported reproductive period in the Mediterranean for the species (July–October [37,71,72]), while with regards to *S. elongata*, our records (mature females in July and September) have represented the first information on spawning period for its entire geographical distribution.

A longer spawning cycle (November–March) was, indeed, observed for the blue-mouth *H. dactylopterus,* with a peak of actively spawning females having embryos inside the ovaries in January. This period differs from the observations of the western Mediterranean Sea (e.g., [3,8,34]), while it is consistent with data from Portugal and the Azores [73,74,75]. The presence of spermatic crypts found in developing females from August to October (and then from November to March in developing, non- and actively spawning specimens) confirmed a specific reproductive strategy of the species, with a prolonged sperm storage period (8 months) available for period with a low possibility of mating. Ref. [33] also described a delay of 1 to 3 months between insemination and fertilization, and, as suggested by [8,74,75] reported a long sperm storage.

All of the analyzed species were batch-spawners in association with an asynchronous ovarian organization. Indeed, based on the OSFD in the ovaries of sexually mature females, all species were characterized by continuous oogenesis with multiple oocyte deposition. This type of oogenesis was found also in the tropical fishes of the family Scorpaenidae (genera *Scorpaenopsis* and *Scorpaena*), and appears to be associated, in the literature, with small GSI values in the females with mature ovaries (1.08–1.60% in *Scorpaenopsis papuensis*, 0.89–1.05% in *Sc. possi,* and 1.3–1.61% in *S. cyanostigma* [57,58,76]). Our results show a completely different trend for the studied species with high values of GSI in mature females (among *Scorpaena* species, 15.34% GSI maximum in *S. notata,* and 9.48% maximum GSI in *H. dactylopterus*).

The stable number of cortical alveoli and vitellogenic oocytes observed throughout the spawning season in all the species analyzed indicated a continuous recruitment of oocytes and the occurrence of de novo vitellogenesis (i.e., process of producing vitellogenic oocytes from previtellogenic oocytes during the spawning season, and consequent recruitment into the standing stock of yolked oocytes). This result has been already observed in *H. dactylopterus* in Spanish and Portuguese waters [75,77], but not in the *Scorpaena* species. The OSFD showed that each cohort of oocyte was at the same stage, and sequential developmental stages co-occurred, ranging from the onset of secondary growth (early cortical alveoli stage) to final oocyte maturation. This type of modal dynamic organization is quite typical for fish populations with an indeterminate fecundity type [49,78]. Indeed, in *H. dactylopterus* and in the four *Scorpaena* species analyzed, the co-occurrence of very early SG oocytes with new POFs indicating that oocyte recruitment was taking place in actively spawning females. In addition, although during the spawning season low atresia was detected, a marked increase in the relative intensity of α-atresia (up to 50%) was found in the ovaries at the end of spawning season. This pattern, already found in *H. dactylopterus* [74,77], has never been observed in the *Scorpaena* species here analysed, thus, it seems to corroborate the hypothesis of an indeterminate fecundity, clearly distinguishing these species from fish characterized by determinate fecundity, in which very low values of incidence of atresia are expected even at the end of the spawning season (e.g., [79,80]). Usually, fish with indeterminate fecundity, such as *Merluccius merluccius* [47], *Trachurus trachurus* [81], and *Scomber scombrus* [82], show a continuous oocyte recruitment throughout the spawning period exhibiting high fraction of early secondary growth oocytes, such as that observed in our species. Currently, the fecundity type designation of *H. dactylopterus* is not unanimous: ref. [77], suggested a determinate fecundity, whereas [75] suggested an indeterminate one. Our observations may support an indeterminate fecundity type for the blue-mouth and for the rockfish species. However, our study lacks either laboratory or field assessment of POF degeneration rate, and needs to be validated with a more precise evaluation of the time-period of full POF resorption.

## 5. Conclusions

Information reported here represents a step towards the expansion of the knowledge concerning the reproductive strategies of the ovuliparous species belonging to *Scorpaena* genus and the zygoparous *Helicolenus dactylopterus*. The specialized ovarian morphology observed and the histological findings attest to specialized modes of oviparity of these species, strictly related to the viviparity. Since many characteristics of their reproductive biology affecting the reproductive potential are unknown, the assessment ovarian dynamic of such species in Sardinian populations allowed for the determination of the ovarian modality, the oocyte recruitment process of the species, and the establishment, in a preliminary way, of the type of fecundity to better comprehend their reproductive strategy.

## Figures and Tables

**Figure 1 animals-12-01412-f001:**
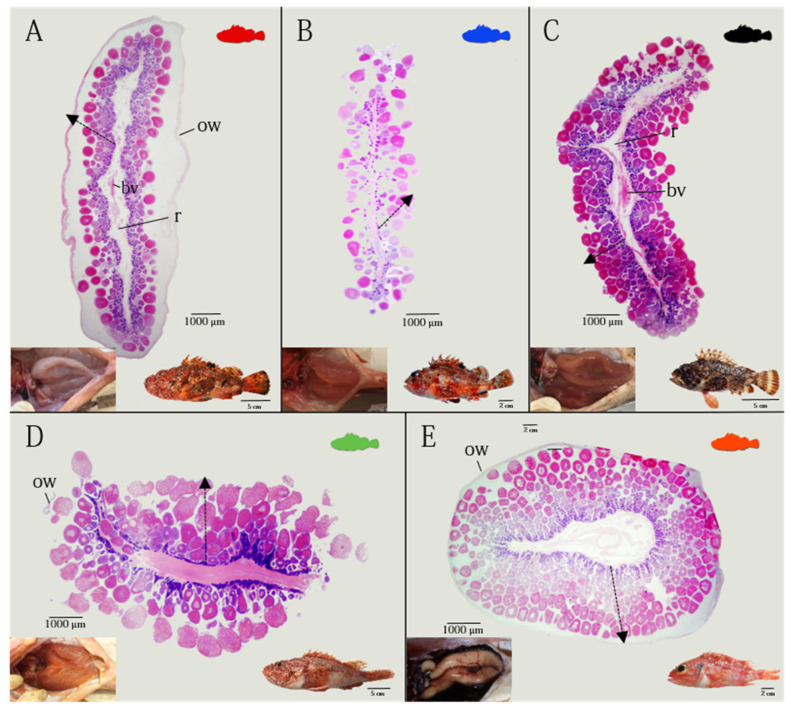
Morphological characteristics of cystovarian type II-3 ovary of the five species studied (H&E). Corresponding macroscopic ovaries, located in the abdominal cavity above the bladder, are also shown down on the left of each species. (**A**) *Scorpaena scrofa* (TL 22.3 cm); (**B**) *Scorpaena notata* (TL 11.3 cm); (**C**) *Scorpaena porcus* (TL 15.6 cm); (**D**) *Scorpaena elongata* (TL 28.7 cm); (**E**) *Helicolenus dactylopterus* (TL 18.2 cm). bv, blood vessel; ow, ovarian wall; r, rachis. The dashed line shows the radial development of oocytes from the stroma to surrounding ovarian wall.

**Figure 2 animals-12-01412-f002:**
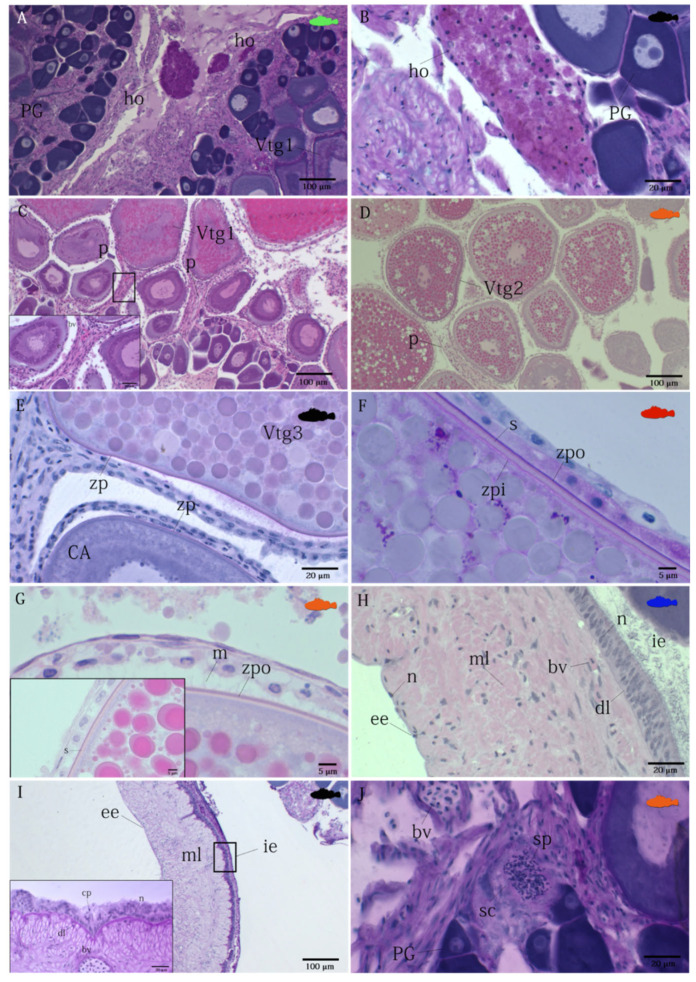
Photomicrographs of ovarian histology, depicting: (**A**) *Scorpaena elongata*, cross section of a maturing ovary in which holocrine glands, reacting positively to PAS staining, are present (AB/PAS). (**B**) *Scorpaena porcus*, high magnification of an holocrine gland PAS-positive (AB/PAS); (**C**) Scorpaena scrofa, cross-section of a mature/spawning ovary, in which ovarian follicles are connected to the stroma by a branching, vascularized peduncle (H&E); (**D**) *Helicolenus dactylopterus*, a long peduncle which encompasses the follicular cells of its vitellogenic oocytes (H&E); (**E**) *Scorpaena porcus*, oocytes at different maturity stages (CA and Vtg3), where diverse thickness of the zona pellucida, strongly PAS-positive, are evident (AB/PAS); (**F**) *Scorpaena scrofa*, high magnification of inner and outer layers of zona pellucida in a vitellogenic oocyte. Striations are also visible (AB/PAS). (**G**) *Helicolenus dactylopterus*, outer and inner zona pellucida with microvilli in evidence. Striations in great detail (H&E). (**H***) Scorpaena notata*, cross-section of an ovarian wall during developing stage (H&E); (**I**) Scorpaena porcus, cross-section of the ovarian wall during spawning stage, where active secretory epithelium is visible in greater detail (AB/PAS). (**J**) *Helicolenus dactylopterus*, spermatic crypts packed with spermatozoa. PAS-positive granules are visible, realized by the epithelium (AB/PAS). bv, blood vessel; CA, cortical alveoli; cp, cytoplasmatic projections; dl, dense line; ee, external epithelium; ho, holocrine gland; ie, internal epiuthelium; m, microvilli; ml, muscular layer; n, nucleus; p, peduncle; PG, primary growth oocyte; s, striation; sc, spermatic crypt; sp, spermatozoa; Vtg1, primary vitellogenic oocyte; Vtg2, secondary vitellogenic oocyte; Vtg3, tertiary vitellogenic oocyte; zp, zona pellucida; zpi, inner zona pellucida; zpo, outer zona pellucida.

**Figure 3 animals-12-01412-f003:**
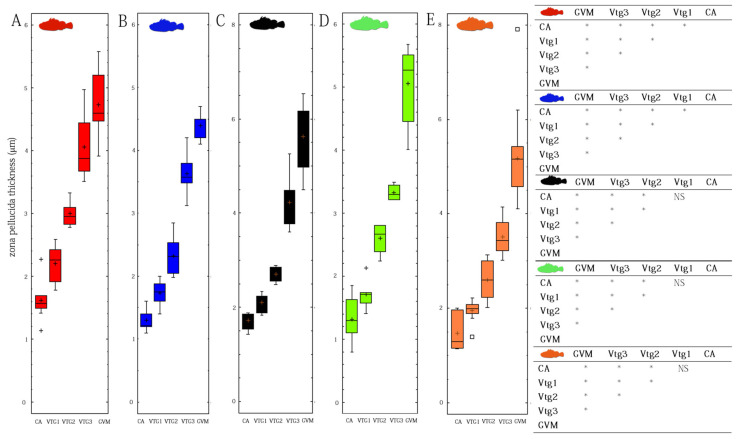
Box- and whisker-plots showing mean value of zona pellucida thickness at different maturity stages. The box represents the 25th and 75th quantile. The points beyond the whiskers are potential outliers. (**A**) Scorpaena scrofa; (**B**) *Scorpaena notata*; (**C**) *Scorpaena porcus*; (**D**) *Scorpaena elongata*; (**E**) *Helicolenus dactylopterus*. CA, cortical alveoli; GVM, germinal vesicle migration; VTG1, primary vitellogenic; VTG2, secondary vitellogenic; VTG3, tertiary vitellogenic. For each species, a table with significant differences (*p*-value < 0.05, *) between maturity stages is provided (NS, not significant differences).

**Figure 4 animals-12-01412-f004:**
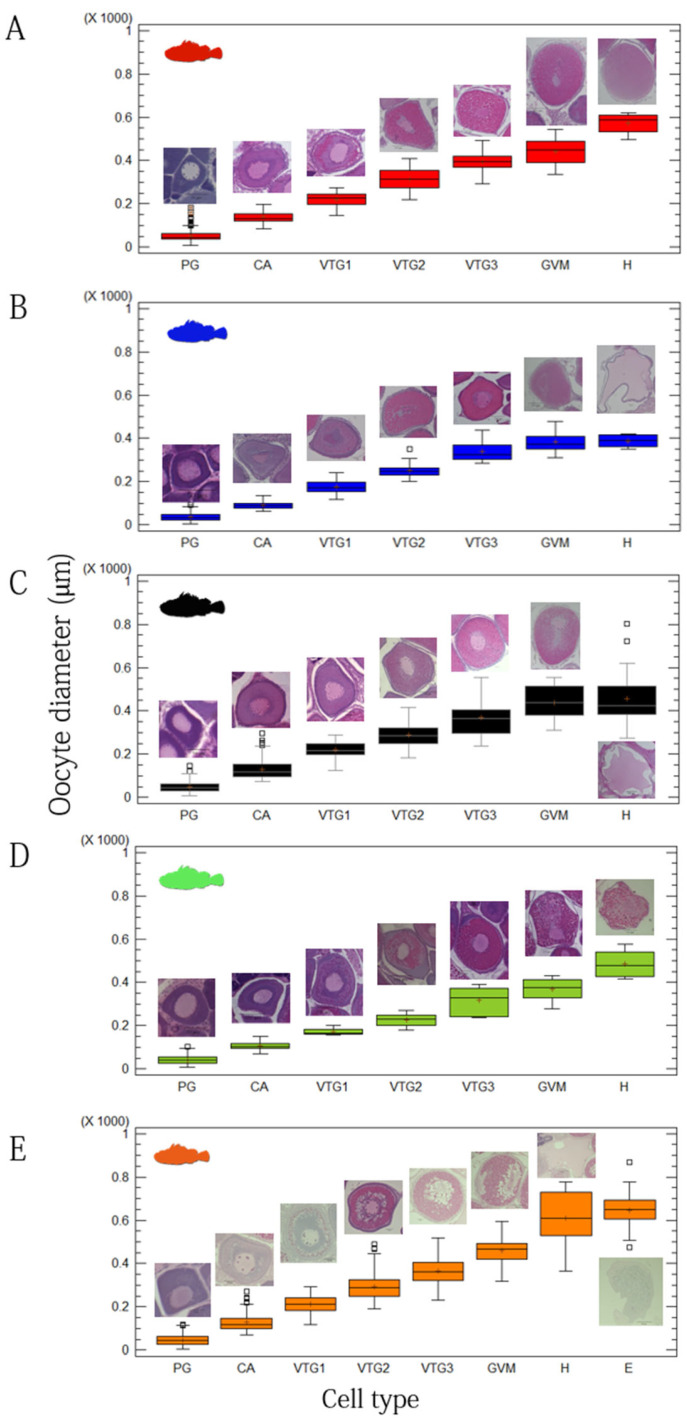
Box- and whisker-plots showing the mean oocyte dimension (diameter, µm) of different cell types in the five studied species. (**A**) *Scorpaena scrofa*, number of measured oocytes = 1888; (**B**) *Scorpaena notata*, number of measured oocytes = 1901; (**C**) *Scorpaena porcus*, number of measured oocytes = 2401; (**D**) *Scorpaena elongata*, number of measured oocytes = 1600; (**E**) *Helicolenus dactylopterus*, number of measured oocytes = 3140. The box represents the 25th and 75th quantile. The points beyond the whiskers are potential outliers. Above or under each box, the relative image of the cell type for each species investigated is shown.

**Figure 5 animals-12-01412-f005:**
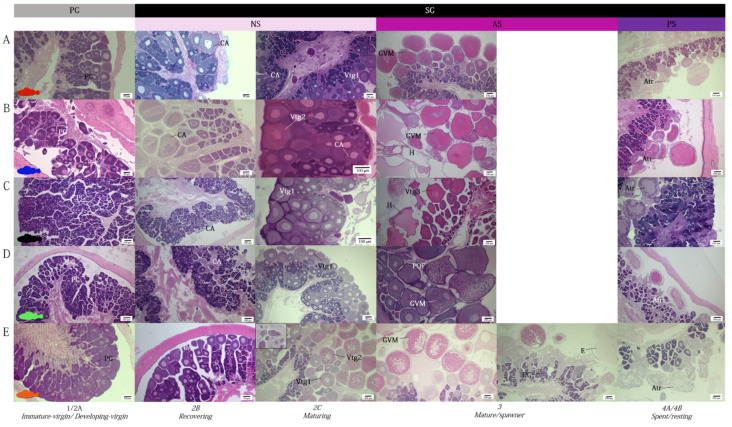
Female developmental maturity stages of the five studied species. (**A**) *Scorpaena scrofa*; (**B**) *Scorpaena notata*; (**C**) *Scorpaena porcus*; (**D**) *Scorpaena elongata*; (**E**) *Helicolenus dactylopterus*. The Primary growth phase (PG) included immature virgin and virgin-developing ovaries, while the multiple secondary growth (SG) phase included non-spawning (NS) females (with no spawning markers, recovering and maturing females), actively spawning (AS) females (with markers of imminent or recent spawning activity, spawner/mature females) and post-spawning (PS) females (a great amount of atretic oocytes are present, spent, and resting females). Atr, Atretic oocyte; CA, cortical alveoli; E, embryo; GVM, germinal vesicle migration; H, hydration; PG, primary growth oocyte; POF, post-ovulatory follicle Vtg1, primary vitellogenic; Vtg2, secondary vitellogenic; Vtg3, tertiary vitellogenic.

**Figure 6 animals-12-01412-f006:**
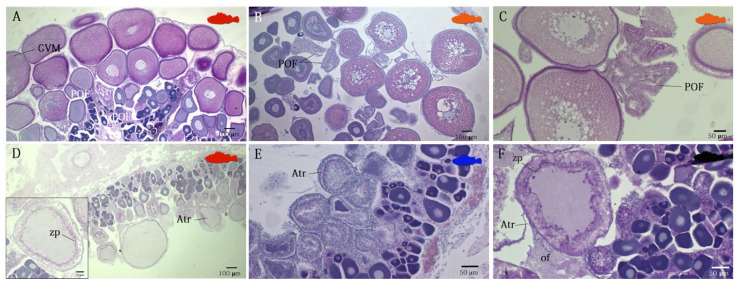
Post-ovulatory follicles and atretic oocytes in *Scorpaena* genus and in *Helicolenus dactylopterus* (AB/PAS staining). (**A**) Actively spawning females of *S. scrofa*, (**B**) *H. dactylopterus* and (**C**) *S. elongata* that have already spawned for the presence of new post-ovarian follicles. Atretic oocytes with irregular contour and zona pellucida which are PAS-positive and with partially liquefied yolk globules at the end of spawning in (**D**) *Scorpaena scrofa*, (**E**) *Scorpaena notata,* and (**F**) *Scorpaena porcus.* Atr, atretic oocyte; GVM, germinal vesicle migration stage; of, ovarian fluid; POF, post-ovulatory follicle; zp, zona pellucida.

**Figure 7 animals-12-01412-f007:**
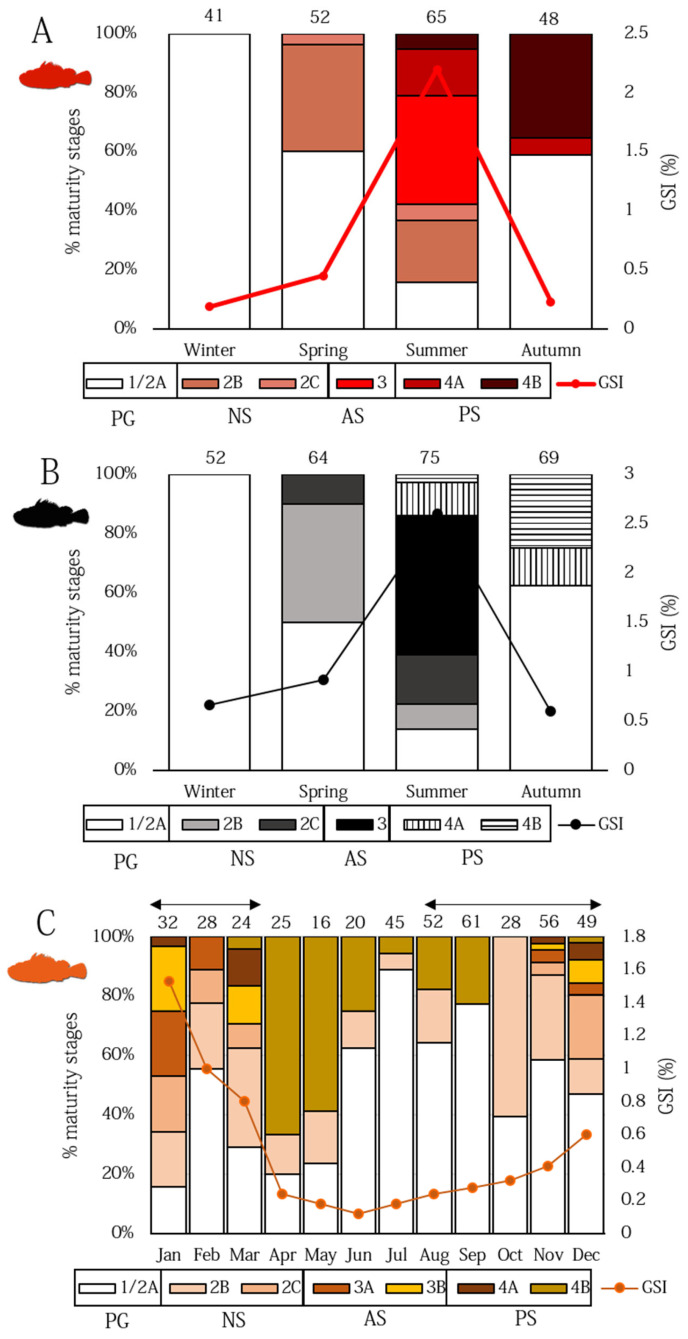
Proportions of *Scorpaena scrofa* (**A**), *Scorpaena porcus* (**B**), and *Helicolenus dactylopterus* (**C**) females in relation to their gonadal growth stage (PG, primary growth; NS, non-spawning; AS, actively spawning; PS, post-spawning). *S. scrofa* and *S. porcus* were described per season, while *H. dactylopterus* per month. The evolution of the gonado–somatic index (GSI (%)) is also represented. The double-headed arrow in Figure 7C indicates the presence of spermatic crypts inside the ovaries. Above each season or month, the number of analyzed females is reported.

**Figure 8 animals-12-01412-f008:**
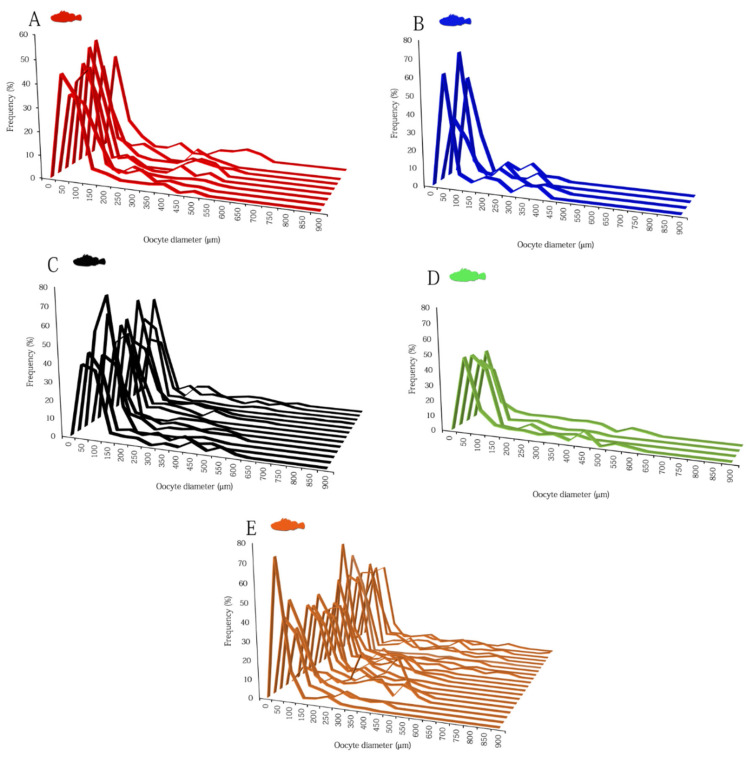
Oocyte size frequency (OSFD) distributions from 8 *Scorpaena scrofa* (**A**), 4 *Scorpaena notata* (**B**), 13 *Scorpaena porcus* (**C**), 4 *Scorpaena elongata* (**D**), and 20 *Helicolenus dactylopterus* (**E**) females.

**Figure 9 animals-12-01412-f009:**
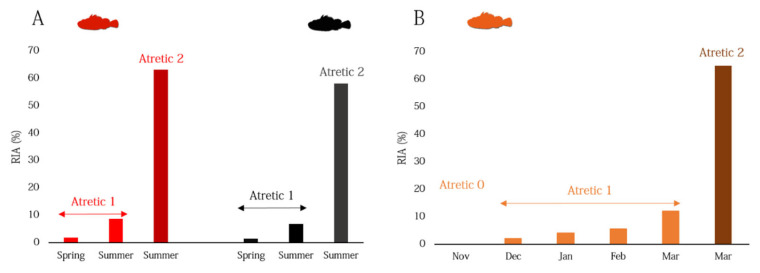
Relative intensity of atresia (RIA) through the spawning season in females of *Scorpaena scrofa* and *S. porcus* (**A**) and *Helicolenus dactylopterus* (**B**). Atretic 0: ovaries with 0% of SG oocytes with α-atresia; Atretic 1: ovaries with <50% of SG oocytes with α-atresia; Atretic 2: ovaries having ≥50% SG oocytes with α-atresia. In *Scorpaena* species date are expressed per season, while in *H. dactylopterus* per month.

**Table 1 animals-12-01412-t001:** List of the selected species: number of collected females (N), size (TL), and depth range at which the individuals were caught are presented.

Species	N	Size Range (TL, cm)	Depth Range (m)
Family Scorpaenidae			
*Scorpaena elongata* Cadenat, 1943	45	19.5–47.5	60–478
*Scorpaena notata* Rafinesque, 1810	33	8.1–14.2	13–101
*Scorpaena porcus* Linnaeus, 1758	260	9.1–21.6	21–155
*Scorpaena scrofa* Linnaeus, 1758	206	9.5–36.9	30–155
Family Sebastidae			
*Helicolenus dactylopterus* (Delaroche, 1809)	436	7.8–34.2	60–651

**Table 2 animals-12-01412-t002:** Histochemical properties of the internal epithelium of the ovarian wall, ovarian fluid, and holocrine gland in the rachis in *Helicolenus dactylopterus* (HD), *Scorpaena elongata* (SE); *Scorpaena notata* (SN), *Scorpaena porcus* (SP) and *Scorpaena scrofa* (SS). PAS, periodic acid–Schiff; AB, Alcian blue; +, presence; −, absence.

Structures	PAS					AB pH 2.5				
	HD	SE	SN	SP	SS	HD	SE	SN	SP	SS
Ovarian wall	+	+	+	+	+	+	+	+	+	+
Ovarian fluid	+	+	+	+	+	−	−	−	−	−
Holocrine gland	+	+	+	+	+	−	−	−	−	−

## Data Availability

The data presented in this study are available on request from the first author.

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
