# Peer review of "Investigating the Ovarian Microstructure in the Genera Helicolenus and Scorpaena (Teleostei, Sub-Order Scorpaenoidei) with Implications for Ovarian Dynamics and Spawning"

_animals, 2022, doi:10.3390/ani12111412_

Round 1

Reviewer 1 Report

  1. figure1 lacks A, B, C , and in the caption,line 218 lacks (E)
  2. why  is there no ovarian wall structures showed in figure B,C,D, since all of them have ovarian wall said in table 2.
  3. please add these papers in references: 

    New insights on folliculogenesis and follicular placentation in marine viviparous fish black rockfish (Sebastes schlegelii). Gene. 2022, 827: 146444.

    Multiple fetal nutritional patterns before parturition in viviparous fishSebastes schlegelii (Hilgendorf, 1880). Frontiers in Marine Science, 2021, 7: 571946.Sperm maturation, migration, and localization before and after copulation in black rockfish (Sebastes schlegelii). Theriogenology, 2021, 166, 83-89.

Reviewer 2 Report

The present study described the ovarian morphology with a histological and histochemical approach, in four ovuliparous species belonging to Scorpaena genus compared with a zygoparous species. The design of this paper is fine and the results are solid. However, there are some minor corrections need to be made before publication.

1line 29: “aimsin” should be replaced with “aims in”.

2line 76: “The ovarian type corresponds to the phylogenetic classification”, no references were listed. A tree of phylogenetic classification associated with ovarian types should be presented in this study.

3line 78: The cystovarian type includes four variant types, what are the other two

4line 93: The reproductive mode of five black rockfish in this study should be presented.

5line 136: The meaning of “Depth range” should be presented in the table 1 in this study.

6line 144: The seven stages of all females correspond to the developmental stages of oocytes in this study which should be presented in this study.

7line 166: The vitellogenic stages, Vtg1, Vtg2, Vtg3, what are the criteria to distinguish them?

8line 172: Why use α-atresia to determine to atretic states? Only one type follicular atresia in five black rockfish in this study? β-atresia?

9line 184: What are maturity stages in developmental stages of oocytes? I think it is not really accurate in here.

10line 207: What is ovarian capsule? Does it refer to ovarian wall? Proper nouns should be unified in this study.

11line 210: The structure of ovarian lamellae, rachis, ovarian wall should be marked clearly in the figure 1 in this study.

12line 213: what is pelagic gelatinous matrix? Does it refer to connective tissues surrounded by oocytes? There is no such reference in previous studies.

13line 214: The size of fish body should have a scale as a reference in the figure 1 in this study to present the total length, and the diagram of ovarian appearance are not clear, the “bv” should have a maginified picture, the figture notes and the figture 1C and 1D are non-corresponding, lacking of figure 1E.

14line 225: The length of vascularized peduncle is increased in vitellogenic stages which are not be presented in the figure 2D.

15line 227: No figure 5F in this study.

16line 229: A high magnification of zona pellucida should be presented in the figure 2E, 2F, 2G in this study to show thickness, the structure of three layers and multilamellar striated region.

17line 240: “the period of spawning” should be replaced with “the developmental stage of oocyte” in detail.

18line 248: The ovary is immature in figure 2A, the developmental stage of ovary in figure 2C should be presented in this study.

19line 252255: The latin “Scorpaena scrofa” “Scorpaena porcus” should be italicized.

20line 257: The structure of striations needs a magnified picture.

21line 260: Figure 2F should be replaced with figure 2I.

22line 276: The figure 3 don’t mark with significant difference.

23line 286: The figure 4 don’t mark with significant difference.

24line 309: The hydrated oocytes should be marked in the figure 6A-C.

25line 333: How many individuals are counted in each quarter or month that are not displayed?

Reviewer 3 Report

Minor revisions required:

Line 29 = aimsin =aims in

Line 38 – “de novo vitellogenesis”? In the Discussion, can a brief statement be added what is meant by this?

 Line 161 = Ovarian dynamic? Dynamics?

 Line 202 – parallely=parallel

 Enlarge or bolden the labels on the photos in Figure 2, or change colour to improve contrast. This comment also applies for Figure 5 and 6 where the labels are even more difficult to see.

Overall, the manuscript provides detailed histological description of the unique reproductive characteristics of the species of fish studied. The data presented are valuable not only for understanding species biology but also for culture and conservation purposes. The authors are commended for their patience in collecting and analysing the data. 
